# Multi-Modality Brain Disease Prediction with Progressive Curriculum Graph Learning

## Abstract

Recently, graph-based multi-modal learning approaches have been studied to handle multi-modal medical brain data analysis. Although they have achieved some promising performance, they still suffer from two main issues. *First*, current works generally fail to capture the inherent relationships of subjects (samples) from both feature and semantic/label perspectives. *Second*, for brain disease prediction tasks, the number of modalities is usually large (usually more than 4) and existing methods generally employ simple multi-modal fusion techniques that fail to carefully capture the dependencies of different modalities. To address these issues, this paper proposes a novel approach for multi-modal brain disease prediction by developing curriculum multi-modality learning. Our approach stems from observing that multi-modality learning becomes more challenging as the number of modalities increases, while recognizing curriculum learning as providing an explicit mechanism for tackling easy-to-hard learning tasks. This motivates us to propose a new progressive multi-modality learning strategy by **leveraging the curriculum learning pipeline**. Specifically, we first propose to dynamically learn a context-graph representation by jointly modeling the relationships of subjects from both feature and semantic label cues. Then, we propose a new multi-modality brain data representation by employing progressive curriculum learning. Experiments demonstrate the advantages of the proposed curriculum multi-modal learning strategy. The code of our method will be released upon acceptance.

## 1 Introduction

Brain disease prediction has gained increasing attention in the field of computer aided diagnosis in recent years. Some works have been proposed for Autism Spectrum Disorder (ASD) and Alzheimer's Disease (AD) prediction (Pan et al., 2021; Baydargil et al., 2021; Duan et al., 2023; Wen et al., 2022; Zhang et al., 2024). For example, Pan et al. (Pan et al., 2021) propose a Multi-view Separable Pyramid Network (MiSePyNet) to excavate information on a single modality and improve the early diagnosis of Mild Cognitive Impairment (MCI) and Alzheimer's Disease (AD). Singh et al. (Singh & Kumar, 2024) propose a pre-trained convolutional neural network for Magnetic Resonance Imaging (MRI) data to recognize the different stages of AD. Jiang et al. (Jiang et al., 2020) propose a hierarchical graph convolution network for disease prediction by jointly considering the brain network information and the association between subjects.

However, the above methods are generally developed for brain disease prediction with single-modality data. In practical application, multi-modality data can be generated from different sources such as Positron Emission Tomography (PET), Magnetic Resonance Imaging (MRI), clinical data, and biomarkers (Zheng et al., 2022; Song et al., 2023). Some previous works demonstrate the effectiveness of multi-modality learning on enhancing the model's performance by considering the consistency and complementary of different modalities (Zheng et al., 2022; Feng et al., 2023; Zhang et al., 2023a). Therefore, recent studies mainly focus on disease prediction by using multi-modality data (Lu et al., 2023; Yang et al., 2023; Zhao et al., 2024). For example, Zhang et al. (Zhang et al., 2023a) propose a novel Multi-modal Cross-Attention Diagnostic (MCAD) model to achieve the interaction of multi-modal data for AD diagnosis. Yang et al. (Yang et al., 2023) propose a multi-model fusion framework by developing an adaptive fusion module for multi-modal AD data. Zheng et al. (Zheng et al., 2022) employ a cross-modal attention mechanism to encode the interaction and dependency between modalities and thus improve the prediction performance. Song et al. (Song

et al., 2023) propose a dual-modality fused brain connectivity network and introduce multi-center attention into a graph convolutional network for AD diagnosis.

Inspired by the powerful expressive capability of graphs, some recent works also develop graph-based methods for multi-modality brain data learning. For instance, Huang et al. (Huang & Chung, 2020) propose the Edge-Variational GCN (EV-GCN) framework which adaptively fuses subjects' multi-modal data into a global feature graph for uncertainty-aware disease prediction. Kazi et al. (Kazi et al., 2019b) propose a new spectral domain method (InceptionGCN) to capture the feature associations of subjects and Region of Interests (ROIs) in the brain. Zheng et al. (Zheng et al., 2022) propose a novel Multi-Modal Graph Learning (MMGL) framework which deals with multi-modal data to construct a global graph from feature perspective for brain disease prediction. Liu et al. (Liu et al., 2025) propose a novel Multi-modal Multi-Kernel Graph Learning (MMKGL) method to address the negative impact of multi-modal integration and extract the heterogeneous information from feature graphs with different modalities. Notably, MMKGL (Liu et al., 2025) only utilizes label information to constrain feature representations of subjects without further considering the semantic information brought by labels. In summary, the existing works generally suffer from several issues. First, existing works generally construct a feature graph and ignore fully modeling the relationships of subjects (samples) from both feature and label perspectives. Second, for brain disease prediction, the number of modalities is usually large (usually more than 4). As the number of modalities increases, the training data becomes more complex, rendering the task of multi-modal learning increasingly challenging. However, current works generally adopt the simple multi-modal fusion which fail to carefully capture the complementary information across different modalities.

To address these issues, in this paper, we propose a novel graph based multi-modality learning approach for brain disease prediction tasks. Our proposed approach contains two main aspects. First, we propose to dynamically learn a context-graph representation by jointly encoding both feature and semantic label relationships of subjects to learn rich context-aware feature representation for each subject. Second, with growing numbers of modalities employed in brain disease prediction, training data becomes progressively more complex, causing the challenge of multi-modal learning to escalate accordingly. It is known that curriculum learning provides an explicit scheme for the easy-to-hard learning task. This motivates us to develop a new progressive multi-modality learning approach by leveraging the curriculum learning pipeline. To our best knowledge, this paper is the first work to exploit curriculum learning into multi-modality fusion learning tasks. Specifically, inspired by work (Yang et al., 2022), we introduce the curriculum supervision for multi-modality learning strategy by designing label-smoothing based easy-to-hard supervisions which are coupled with single-to-many modalities and thus can gradually capture the complementary information of multiple modalities. The whole process is trained in an end-to-end way. On two commonly utilized datasets, our method outperforms other advanced methods and achieves a new state-of-the-art performance. Overall, the main contributions of this paper are summarized as follows:

- We propose to exploit a new progressive curriculum learning for multi-modality data learning. The method employs the curriculum supervision to capture the complementary information of different modalities very effectively for comprehensive data representation. Note that, the proposed pipeline is a general scheme which can be applied for various multi-modality learning with many modalities.

- We propose a new dynamical context-graph representation that jointly exploits both feature and semantic relationships of subjects to learn context-aware feature representation for each subject/sample.

- Experimental results on two public benchmarks demonstrate that the proposed method can achieve a new state-of-the-art performance on multi-modality brain disease prediction tasks, validating the effectiveness of the proposed dynamical context graph model and multi-modal curriculum learning pipeline.

## 2 RELATED WORKS

### 2.1 MULTI-MODALITY BRAIN DISEASE PREDICTION

In recent years, some works have been developed for multi-modality brain disease prediction (Parisot et al., 2018; Zheng et al., 2022; Hao et al., 2023). Among them, the graph-based methods gain

impressive abilities to deal with multi-modality medical data (Parisot et al., 2017; Zheng et al., 2022; Liu et al., 2025). For example, PopGCN (Parisot et al., 2017) first introduces Graph Convolutional Network (GCN) for Autism Spectrum Disorder and Alzheimer's Disease prediction. InceptionGCN (Kazi et al., 2019b) proposes a spectral domain architecture for disease prediction to capture the structural heterogeneity of intra-graph and inter-graph. Latent-Graph Learning (LGL) (Cosmo et al., 2020) learns a graph based on multi-modality features by dynamic graph pruning and presents an end-to-end trainable network for disease prediction. MMGL (Zheng et al., 2022) proposes a modality-aware representation learning to capture the relationship between modalities and enhance the brain disease prediction. MCAD (Zhang et al., 2023a) introduces a multi-modal interaction module to integrate the information of different modalities by using cross-attention mechanism. MMKGL (Liu et al., 2025) presents a multi-modal graph embedding module to adaptively generate multiple graphs for multiple modalities and further introduces function and supervision graphs to guide the multi-graph fusion. Different from existing works, in this paper, we first learn a new context-graph for multi-modality brain data by building a **dual-relation graph** to model both feature and label correlations of subjects. Then, we design a specific dual graph convolutional network module to achieve context-graph representation and learning (§3.2).

## 2.2 CURRICULUM LEARNING FOR MULTI-MODALITY LEARNING

As an important machine learning approach, curriculum learning has been applied on some multi-modality learning tasks. For instance, Gong (Gong, 2017) proposes a novel multi-modal curriculum learning algorithm for label propagation which first investigates the difficulty of unlabeled samples from multiple modalities and then optimizes the propagation sequence in an easy-to-hard way. Xu et al. (Xu et al., 2018) propose a Multi-modal Self-paced Learning (MSPL) method for image classification to learn the easy to difficult knowledge from both sample and modality level. Zhou et al. (Zhou et al., 2023) propose a Intra-Modal and Inter-Modal Curriculum Learning (I2MCL) framework to simultaneously consider both data difficulty and modality balance for multi-modality learning. Zhang et al. (Zhang et al., 2023b) propose a novel Hypersphere-based Visual Semantic Alignment (HVSA) network via curriculum learning to align remote sensing image-text pairs from the easy-to-hard manner.

Our method is significantly different from the above existing works. Previous works generally exploit curriculum learning to address the difficult data samples. In contrast, we exploit the curriculum graph learning strategy to achieve **progressive multi-modal interaction** by gradually capturing the complementary information between different modalities, as presented in §3.3. To our best knowledge, this paper is the first work to exploit curriculum graph learning for multi-modality interaction learning.

## 3 THE PROPOSED METHOD

In this section, we mainly propose a novel unified framework for multi-modality brain disease prediction, which integrates the curriculum learning with dual graph convolution to better serve the feature representation learning within and between modalities. The whole framework is shown in Figure 1. Below, we will introduce the following four modules in detail:

- Initial Feature Selection: To obtain better initial features, we first select features based on the morphological features of each modality to obtain complete multi-modality data, as conducted in works (Parisot et al., 2017; Zheng et al., 2022).

- Context-Graph Learning: We develop Context-Graph Learning module to capture the relationships of intra-modality subjects by encoding both feature and label relationships of subjects.

- Dual Graph Convolutional Module: We leverage a Dual Graph Convolutional Module to conduct message passing over context-graph and achieve the information interaction between subjects with different modalities.

- Multi-Modality Curriculum Learning: To model the dependencies of modalities, we introduce the Curriculum Learning to progressively conduct modal interaction between different modalities.

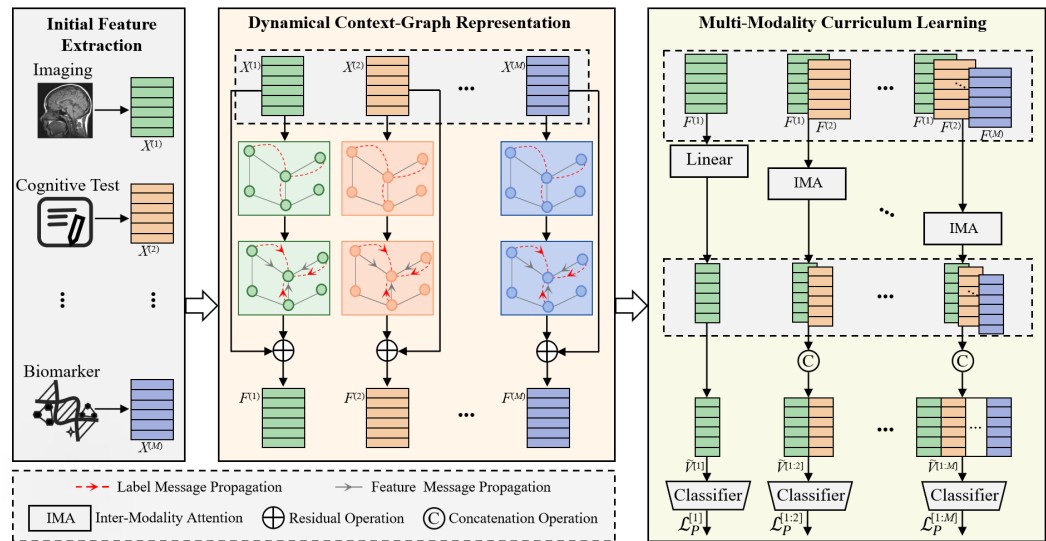

Figure 1: The overall network structure of our proposed method which includes Initial Feature Extraction, Dynamical Context-Graph Representation and Multi-Modality Curriculum Learning. First, the initial features of each modality are obtained through feature preprocessing. Second, the Dynamical Context-Graph Representation is used to obtain richer feature representation of each modality. Finally, the fusion features are generated with Multi-Modality Curriculum Learning and the fused features are fed into the classifiers to get the disease prediction results.

## 3.1 INITIAL FEATURE EXTRACTION

We leverage two widely utilized open brain disease datasets, namely TADPOLE and ABIDE datasets. For the TADPOLE dataset, we follow the previous research (Zheng et al., 2022) to conduct feature selection based on morphological features. Subsequently, we determine whether to retain a subject by calculating the feature loss of each subject. As a result, we acquire relatively comprehensive multi-modality data features. For the ABIDE dataset, we use the Preprocessed Connectomes Project (PCP) (Craddock et al., 2013; Shehzad et al., 2015) to obtain multi-modality features, as conducted in work (Zheng et al., 2022).

## 3.2 DYNAMICAL CONTEXT-GRAPH REPRESENTATION

### 3.2.1 CONTEXT-GRAPH LEARNING

To fully encode the relationships of subjects, we build a dual-relational graph $G(V^{(m)}, E^{(m)}, \hat{E})$ for the $m$-th modality where $m = 1, 2 \cdots M$. It refers to the feature and semantic label branch, encoding both feature and label relationships of different subjects, as shown in Figure 1. Note that, since the label information of each subject is invariant in different modalities, the label correlations $\hat{E}$ are commonly shared in different modalities.

**Nodes**: For the $m$-th modality, the nodes $V^{(m)}$ of our context-graph are represented as the set of subjects. Specifically, we use $X^{(m)} = \{X_1^{(m)}, X_2^{(m)} \cdots X_N^{(m)}\} \in \mathbb{R}^{N \times d_m}$ to denote the collection of all subject's features where $X_i^{(m)}$ denotes the feature representation of the $i$-th subject node in the $m$-th modality. $d_m$ denotes the feature dimension of the $m$-th modality and $N$ denotes the total number of subject nodes.

**Edges**: As introduced before, the edges of our context-graph involve two types, i.e., feature relationships $E^{(m)}$ and label correlations $\hat{E}$ among subjects. We use an adjacency matrix $A^{(m)}$ to represent feature-type relationships in each modality. It is learned dynamically by computing the cosine similarity metric regularized by some specific constraints. Specifically, we first compute $A^{(m)} \in \mathbb{R}^{N \times N}$

as follows,

$$A_{ij}^{(m)} = \cos\big(X_i^{(m)} W^{(m)}, X_j^{(m)} W^{(m)}\big), \tag{1}$$

where $W^{(m)}$ denotes the learnable transformation matrix. Moreover, to learn a more meaningful graph, we first utilize the Dirichlet energy function (Zhou et al., 2004; Jin et al., 2020) to regularize the learned graph as follows,

$$\mathcal{L}_s(A^{(m)}) = \frac{1}{2N^2} \sum_{i,j=1}^{N} A_{ij}^{(m)} \| X_i^{(m)} - X_j^{(m)} \|^2, \tag{2}$$

where $\| \cdot \|$ represents the Frobenius norm. Then, to overcome the possible trivial solution issue and balance the sparse property, we further employ the regularization term as

$$\mathcal{R}(A^{(m)}) = -\frac{\lambda_1}{N} \mathbf{1}^{\mathbf{T}} \log(A^{(m)} \mathbf{1}) + \frac{\lambda_2}{N^2} \| A^{(m)} \|_F^2, \tag{3}$$

where $\mathbf{1} \in \mathbb{R}^N$ represents an all-one vector. $\lambda_1$ and $\lambda_2$ are two hyper-parameters. Therefore, the total loss for learning our context-graph is defined as follows,

$$\mathcal{L}_G(\mathcal{A}) = \frac{1}{M} \sum_{m=1}^{M} \mathcal{L}_s(A^{(m)}) + \mu \cdot \mathcal{R}(A^{(m)}), \tag{4}$$

where $\mu$ denotes the balancing parameter.

Furthermore, in addition to the feature relationships $A^{(m)}$ for each modality, we also represent the label correlations of subjects via label-type edge set $\hat{E}$. Since the label correlations of subjects are invariant in different modalities, we can utilize a single adjacency matrix $\hat{A}$ to encode them as,

$$\hat{A}_{ij} = \begin{cases} 1, & \text{if } Y_i = Y_j \\ 0, & \text{otherwise} \end{cases}, \tag{5}$$

where $Y_i$ and $Y_j$ represent the ground-truth labels for the subject node $i$ and $j$ respectively.

### 3.2.2 Dual Graph Convolutional Module

Based on the above learned context-graph representation $G(X^{(m)}, A^{(m)}, \hat{A})$, we design a Dual Graph Convolutional Network (DGCN) to learn a context-aware representation for each subject. Specifically, similar to GCN (Kipf & Welling, 2017), DGCN adopts two branches in which the layer-wise message propagation in each branch is defined as follows,

$$\begin{aligned}
H_f^{(m,l+1)} &= \text{ReLU}\big[ D^{(m)^{-\frac{1}{2}}} A^{(m)} D^{(m)^{-\frac{1}{2}}} H_f^{(m,l)} \Theta^{(m,l)} \big], \\
H_l^{(m,l+1)} &= \text{ReLU}\big[ \hat{D}^{-\frac{1}{2}} \hat{A} \hat{D}^{-\frac{1}{2}} H_l^{(m,l)} \Theta^{(m,l)} \big], \\
\tilde{H}^{(m,l+1)} &= H_f^{(m,l+1)} + H_l^{(m,l+1)},
\end{aligned} \tag{6}$$

where $l = 0, 1 \cdots L - 1$ and $m = 1, 2 \cdots M$. $D$ is a diagonal matrix with $D_{ii} = \sum_j A_{ij}$. The output $\tilde{H}^{(m,l+1)}$ represents the output hidden features of the $l$-th layer for the $m$-th modality with $H_f^{(m,0)} = H_l^{(m,0)} = X^{(m)}$. $\Theta^{(m,l)}$ denotes the trainable weight parameters which are shared across two branches. Finally, we leverage the residual operation to incorporate the initial features for each modality as follows,

$$F^{(m)} = \tilde{H}^{(m,L)} + \beta \cdot X^{(m)}, \tag{7}$$

where $\beta$ balances two terms.

### 3.3 Multi-Modality Curriculum Learning

As the modality number increases, multi-modality learning becomes increasingly hard. We present a progressive multi-modality learning approach for disease prediction by leveraging curriculum learning pipeline. The proposed pipeline can conduct modality interaction progressively to fully exploit the complementary information of different modalities.

Overall, as shown in Figure 1, given input $M$-modality data $\mathcal{F}^M = \{F^{(1)}, F^{(2)} \cdots F^{(M)}\}$, our method conducts learning on it via $M$ steps with the $M_t$-th step performing learning on previous $M_t$ modalities $\mathcal{F}^{M_t} = \{F^{(1)} \cdots F^{(M_t)}\}$ where $1 \leq M_t \leq M$. Specifically, in the $M_t$-step, we first develop an Inter-Modality Attention (IMA) module to conduct modality interaction among $M_t$ modalities[1]. It first projects $\mathcal{F}^{M_t}$ into Query $\{Q^{(1)} \cdots Q^{(M_t)}\}$, Key $\{K^{(1)} \cdots K^{(M_t)}\}$ and Value $\{V^{(1)} \cdots V^{(M_t)}\}$ respectively by using different trainable projection matrices. Then, it computes the dependencies between different modalities as,

$$\mathcal{S}_i^{(h,k)} = \frac{\exp\left[(Q_i^{(h)})^T K_i^{(k)}/\tau\right]}{\sum_{k=1}^{M_t} \exp\left[(Q_i^{(h)})^T K_i^{(k)}/\tau\right]}, \tag{8}$$

where $h, k \in \{1, 2 \ldots M_t\}$ and $\tau$ is the scale factor to adjust the hardness of attention (Vaswani et al., 2017). Note that $\mathcal{S}_i \in \mathbb{R}^{M_t \times M_t}$ and $\mathcal{S}_i^{(h,k)}$ can be seen as the contribution of the $k$-th modality to the $h$-th modality for the $i$-th subject. Based on it, one can obtain the enhanced features for each modality by aggregating the message from other modalities as follows,

$$\tilde{V}_i^{(h)} = \sum_{k=1}^{M_t} \mathcal{S}_i^{(h,k)} V_i^{(k)} + V_i^{(h)}. \tag{9}$$

We concatenate the features of the above learned multiple modalities together as,

$$\tilde{V}_i^{[1:M_t]} = \mathrm{Con}\left(\tilde{V}_i^{(1)} \cdots \tilde{V}_i^{(M_t)}\right), \tag{10}$$

where $\mathrm{Con}(\cdot)$ denotes the concatenation operation. The classifier head is used to obtain the label prediction $\tilde{Y}_i^{[1:M_t]}$.

In addition, following previous works (Müller et al., 2019; Yang et al., 2022), we incorporate label smoothing into curriculum training process to enrich model's generalization ability. The smoothed ground-truth label for the $M_t$-th classifier head is defined as follows,

$$Y_i^{\alpha^{[1:M_t]}}(t) = \begin{cases} \alpha, & \text{if } Y_i(t) = 1 \\ (1-\alpha)/(c-1), & \text{otherwise} \end{cases}, \tag{11}$$

where $c$ represents the number of classes and $\alpha^{[1:M_t]} \in [0, 1]$ denotes the smoothness factor. $Y_i \in \mathbb{R}^c$ is the ground-truth label vector of subject $i$. Note that, as the number of current modalities $M_t$ increases, the value of $\alpha^{[1:M_t]}$ also increases and $Y_i^{\alpha^{[1:M_t]}}$ is closer to the one-hot vector $Y_i$. Then, the cross-entropy loss at the $M_t$-th step can be defined as,

$$\mathcal{L}_P^{[1:M_t]} = -\sum_{i=1}^{N} \sum_{t=1}^{c} Y_i^{\alpha^{[1:M_t]}}(t) \log\left(\tilde{Y}_i^{[1:M_t]}(t)\right). \tag{12}$$

Finally, after $M$ steps for modal interacting, we obtain the overall loss to train the whole network as,

$$\mathcal{L} = \mathcal{L}_G(\mathcal{A}) + \sum_{M_t=1}^{M} \gamma_{M_t} \mathcal{L}_P^{[1:M_t]}, \tag{13}$$

where $\gamma_{M_t}$ denotes the balanced hyper-parameter and $\mathcal{L}_G(\mathcal{A})$ denotes the loss of graph learning, as introduced in Eq.(4).

## 4 EXPERIMENT

In this section, we evaluate the effectiveness of the proposed method on two commonly used datasets and compare it with some other related works.

---

[1]We obtain features via a linear layer when $M_t = 1$.

### 4.1 DATASETS AND IMPLEMENTATION

**Datasets.** Similar to the previous work (Zheng et al., 2022), we utilize two processed datasets including ABIDE dataset with four modalities and TADPOLE dataset with six modalities. More details on the datasets can be found in the Appendix.

**Implementation Detail.** For TADPOLE, the dropout rate and learning rate are set to 0 and 0.012, respectively. The number of hidden layers and encoding layers are set to 10 and 2. The number of training epochs is 300. For ABIDE, the dropout rate and learning rate are set 0.35 and 0.0038, respectively. The number of hidden layers and encoder layers are set to 18 and 2. The number of training epochs is set to 450. Besides, the number of attention heads is set to 2 on all datasets.

Table 1: Comparisons with state-of-the-art methods.

|  | TADPOLE | | ABIDE | | | |
|---|---|---|---|---|---|---|
| **METHODS** | **ACC(%)** | **AUC(%)** | **ACC(%)** | **AUC(%)** | **SEN(%)** | **SPE(%)** |
| MLP | 82.28±4.39 | 83.13±3.20 | 75.22±8.06 | 79.30±7.95 | 77.35±9.00 | 75.24±10.9 |
| PopGCN (Parisot et al., 2017) | 82.37±5.10 | 80.71±4.21 | 69.80±3.35 | 70.32±3.90 | 73.35±7.74 | 80.27±6.48 |
| InceptionGCN (Kazi et al., 2019b) | 77.42±1.53 | 81.58±1.31 | 72.69±2.37 | 72.81±1.94 | 80.29±5.10 | 74.41±6.22 |
| Multi-GCN (Kazi et al., 2019a) | 83.50±4.91 | 89.34±5.38 | 69.24±5.90 | 70.04±4.22 | 70.93±4.68 | 74.33±6.07 |
| LSTMGCN (Kazi et al., 2019c) | 83.40±4.11 | 82.42±7.97 | 74.92±7.74 | 74.71±7.92 | 78.57±11.6 | 78.87±7.79 |
| EV-GCN (Huang & Chung, 2020) | 88.51±2.34 | 89.97±2.15 | 85.90±4.47 | 84.72±4.27 | 88.23±7.18 | 79.90±7.37 |
| LGL (Cosmo et al., 2020) | 91.37±2.12 | 93.96±1.45 | 86.40±1.63 | 85.88±1.75 | 86.31±4.52 | 88.42±3.04 |
| MMGL (Zheng et al., 2022) | 92.31±1.73 | 93.91±2.10 | 89.77±2.72 | 89.81±2.56 | 90.32±4.21 | 89.30±6.04 |
| MAFGN (Yang et al., 2023) | 92.80±0.92 | 93.32±2.10 | – | – | – | – |
| MMKGL (Liu et al., 2025) | – | – | 91.08±0.59 | 91.01±0.63 | **91.97±0.64** | 90.05±1.37 |
| MGDR (Jiang et al., 2024) | 93.64±3.90 | 94.89±2.96 | 91.39±2.00 | 91.25±2.07 | 89.33±4.55 | 93.16±3.27 |
| **Ours** | **94.31±2.53** | **94.98±2.57** | **91.84±2.39** | **91.68±2.37** | 89.59±3.47 | **93.78±3.67** |

### 4.2 PERFORMANCE COMPARISONS AND ANALYSIS

**Comparisons with the state-of-the-art methods.** We compare our method with several approaches, including Multilayer Perceptron (MLP), PopGCN (Parisot et al., 2017), InceptionGCN (Kazi et al., 2019b), Multi-GCN (Kazi et al., 2019a), LSTMGCN (Kazi et al., 2019c), EV-GCN (Huang & Chung, 2020), LGL (Cosmo et al., 2020), MMGL (Zheng et al., 2022), MAFGN (Yang et al., 2023), MMKGL (Liu et al., 2025) and MGDR (Jiang et al., 2024). We use 10-fold cross-validation for the reliable comparison and report the results on the metric of Area Under Curve (AUC), accuracy (ACC), specificity (SPE) and sensitivity (SEN), as summarized in Table 1. Here, we can observe that (1) our model obtains better results than traditional single graph based learning methods including InceptionGCN (Kazi et al., 2019b), EV-GCN (Huang & Chung, 2020), LGL (Cosmo et al., 2020) and MAFGN (Yang et al., 2023), which indicates the benefit of the proposed dynamical graph learning approach for multi-modality brain disease prediction task. (2) Our model achieves higher performance than some multi-graph learning-based methods, including Multi-GCN (Kazi et al., 2019a), LSTMGCN (Kazi et al., 2019c), MMKGL (Liu et al., 2025) and MGDR (Jiang et al., 2024), which shows the more effective of the proposed context-graph learning method by simultaneously considering both feature and label cues. (3) Our model outperforms the related work MMGL (Zheng et al., 2022) and MGDR (Jiang et al., 2024) which also adopt cross-attention for modality interaction. This obviously demonstrates the more effectiveness of the proposed multi-modality curriculum learning approach to model the dependencies of modalities and exploit the complementary information of different modalities for rich representation.

**Transductive learning analysis.** By dynamically adjusting the training data, we can evaluate our methods in terms of generalization and robustness, as suggested in work (Zheng et al., 2022). To this end, we gradually increase the training set, varying from 10% to 80%. Note that MMGL (Zheng et al., 2022) holds a leading position and is more representative across both datasets compared to other baselines and MLP serves as a typical representative of traditional methods. Therefore, we have chosen these two methods for comparison. As shown in Figure 2, our method generally achieves better performance with the limited training data, which demonstrates the effectiveness and robustness of the proposed method. Also, we can note that as the ratio of training data increases, our method consistently outperforms the related work MMGL (Zheng et al., 2022) in terms of ACC and AUC. This further demonstrates the generalization and robustness of our model under different ratios of training set.

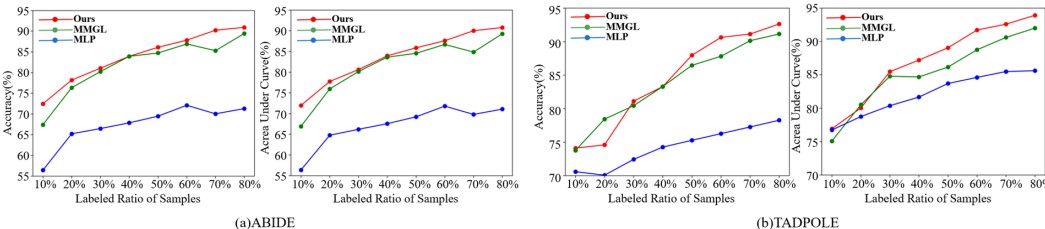

Figure 2: Model performance using different ratios of training samples. Here, we can observe that, the proposed method obtains consistently better performance than the previous MMGL method.

Table 2: Ablation study results on the TADPOLE and ABIDE datasets.

| Components | | | | TADPOLE | | ABIDE | | | |
|---|---|---|---|---|---|---|---|---|---|
| MMCL | LS | FE | LE | ACC(%) | AUC(%) | ACC(%) | AUC(%) | SEN(%) | SPE(%) |
| × | × | × | × | 89.12±3.31 | 90.84±2.63 | 86.48±2.99 | 86.40±3.30 | 85.34±3.57 | 87.45±3.14 |
| ✓ | × | × | × | 92.46±3.57 | 93.21±3.19 | 90.59±2.70 | 90.55±2.71 | 90.07±5.08 | 91.04±4.63 |
| ✓ | ✓ | × | × | 93.47±3.32 | 94.37±3.14 | 91.05±2.94 | 90.98±2.94 | 90.07±5.32 | 91.89±4.23 |
| ✓ | ✓ | ✓ | × | 93.97±2.02 | 94.90±1.78 | 91.62±2.28 | 91.53±2.21 | **90.32±3.24** | 92.73±4.18 |
| ✓ | ✓ | ✓ | ✓ | **94.31±2.53** | **94.98±2.57** | **91.84±2.39** | **91.68±2.37** | 89.59±3.47 | **93.78±3.67** |

## 4.3 ABLATION STUDY

To evaluate the effectiveness of the modules in our approach, we first define a Baseline model by only containing Initial Feature Select (IFS), Inter-Modality Attention (IMA) and Classifier head. In contrast to the baseline, our method consists of four other components including Multi-Modality Curriculum Learning (MMCL), Label Smoothing (LS) strategy (Eq.(11)), Feature-type Edges $E^{(m)}$ (FE) of context-graph and Label-type Edges $\hat{E}$ (LE) of context-graph.

**MMCL.** As we know the task of multi-modality learning becomes increasingly challenging as the number of modalities increases. Note that curriculum learning provides an explicit scheme to address the easy-to-hard learning task so that the model can gradually adapt to the performance of each modality, effectively avoiding the problem caused by direct training of multiple modal data and improving the results of disease prediction. **LS strategy.** To evaluate the feasibility of incorporating smooth labels in modal curriculum learning strategies, we conduct experiments to assess the impact of label smoothing (LS) strategy on model performance. Note that the original real targets are all one-hot, we employ smoothing factors for various modal data. For fewer modalities, a smaller smoothing factor is applied, while conversely, for more modalities, a larger smoothing factor is utilized. This proposed approach aims to prevent the model from making overconfident predictions for modalities with limited data. **FE of context-graph.** We construct a feature graph for each modality, which can better express the feature representation of each modality and also easier to capture the association between the modalities. **LE of context-graph.** Current works generally ignore fully modeling the inherent relationships of subjects (samples) from both feature and semantic/label perspective. We construct a dual graph for each modal data, introducing label-type information into feature information and enriching the information representation of each modality. Not only could the feature representation of modalities be considered, but the label information in each modal representation could also be utilized, as this is beneficial for the model to deeply understand different modal data. Based on it, we further devise a specific dual graph convolutional network to learn context-aware representations for each subject. We study the effect of these components by gradually adding a component into the baseline. Table 2 summarizes the performance. Here, we can observe that the above components can consistently improve the model's performance, which clearly demonstrates the effectiveness of these components in our approach.

In addition, we further investigate the influence of different number of modalities on the model's performance. The evaluation results are shown in Table 3 and 4. In the first row, we proposed the Multi-Modal Course Learning (MMCL) strategy is not utilized. In contrast, the last row illustrates the implementation of MMCL. We can observe that as the modality number increases, the performance of the proposed method is better than the direct fusion multi-modal data method,

Table 3: Comparison results for the different number of modalities on the ABIDE dataset.

| DI | AAQA | AFQA | FMRICN | ACC(%) | AUC(%) |
|---|---|---|---|---|---|
| ✓ | ✓ | ✓ | ✓ | **91.50±3.10** | **91.26±3.11** |
| | | | ✓ | 76.69±3.72 | 76.67±3.69 |
| | | ✓ | ✓ | 89.32±3.92 | 88.97±3.78 |
| | ✓ | ✓ | ✓ | 90.47±3.00 | 90.32±2.99 |
| ✓ | ✓ | ✓ | ✓ | **91.84±2.39** | **91.68±2.37** |

Table 4: Compared results for different number of modalities on the TADPOLE dataset.

| DI | MRI | RF | CFB | CoT | PET | ACC(%) | AUC(%) |
|---|---|---|---|---|---|---|---|
| ✓ | ✓ | ✓ | ✓ | ✓ | ✓ | **93.13±2.57** | **94.10±2.56** |
| | | | | | ✓ | 60.53±3.82 | 74.07±4.19 |
| | | | | ✓ | ✓ | 92.13±1.72 | 93.64±1.84 |
| | | | ✓ | ✓ | ✓ | 93.31±2.90 | 94.39±3.00 |
| | | ✓ | ✓ | ✓ | ✓ | 93.63±3.11 | 94.87±3.15 |
| | ✓ | ✓ | ✓ | ✓ | ✓ | 93.97±2.40 | 94.97±2.46 |
| ✓ | ✓ | ✓ | ✓ | ✓ | ✓ | **94.31±2.53** | **94.98±2.57** |

which demonstrates the advantage of the proposed progressive curriculum learning on exploiting the complementary information of distinct modalities for rich representation.

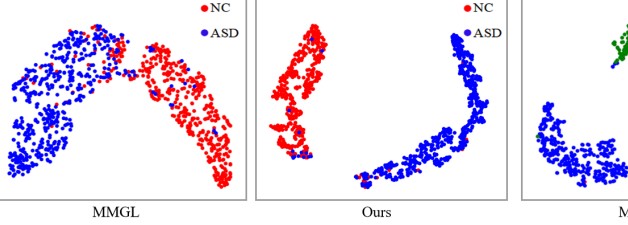

(a) ABIDE  (b) TADPOLE

Figure 3: Comparison of 2D t-SNE visualization for the modality-specific features on two datasets.

## 4.4 VISUALIZATION

To demonstrate the effectiveness of the proposed multi-modality learning, we utilize the 2D t-SNE tool (Kobak & Linderman, 2021) to perform visual comparisons between the proposed method and related work MMGL (Zheng et al., 2022). As shown in Figure 3, comparing with MMGL, our method obtains larger distances between subjects with different classes and maintains smaller distances for the subjects within the same class. This is generally consistent with the prediction performance shown in Table 1 and further demonstrates the effectiveness of our proposed multi-modality method.

## 5 CONCLUSION

In this paper, we propose a new approach for multi-modal brain disease prediction by exploiting dynamical context-graph representation and progressive curriculum learning. The key aspects of the proposed approach are twofold. Firstly, a dynamical context-graph representation that captures both feature and label information of subjects is proposed. Secondly, a new progressive curriculum for learning is leveraged for multi-modality brain data learning. It employs curriculum supervision to capture the complementary information of distinct modalities for rich representation. We evaluate the effectiveness of the proposed method on two public benchmarks (ABIDE and TADPOLE). Experimental results demonstrate that the proposed method can achieve state-of-the-art performance.

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

# A APPENDIX

## A.1 DATASETS

Table 5: Statistics of subjects on the TADPOLE dataset. The Mini-Mental State Examination (MMSE) and the Montreal Cognitive Assessment (MoCA) are widely used instruments for screening cognitive function.

| TADPOLE | Female/Male | Age | MMSE | MoCA |
|---------|-------------|-----|------|------|
| NC | 114/95 | 72.81±5.96 | 29.13±1.11 | 25.93±2.45 |
| MCI | 144/171 | 70.87±7.19 | 28.14±1.70 | 23.53±3.10 |
| AD | 30/44 | 73.29±7.97 | 22.82±2.93 | 16.86±5.06 |

Table 6: Statistics of subjects on the ABIDE dataset.

| ABIDE | Female/Male | Age | Open/Closed-Eye |
|---|---|---|---|
| NC | 90/378 | 16.84±7.23 | 321/147 |
| ASD | 54/349 | 17.07±7.95 | 288/115 |

**TADPOLE**. TADPOLE dataset is a subset of the Alzheimer's Disease Neuroimaging Initiative (ADNI) (Marinescu et al., 2018; 2021; Zheng et al., 2022) which includes both imaging and non-imaging data. After pre-processing, we obtain imaging data containing two modalities, i.e., PET and MRI, where PET offers metabolic information on the subject's tissue, and MRI provides structural information on the subject's tissue. Besides, non-imaging data consists of modalities of Cognitive Tests (CoT), Cerebrospinal Fluid Biomarkers (CFB), Risk Factors (RF), and Demographic Information (DI). This dataset contains 598 subjects, including 209 with Normal Controls (NC), 315 with Mild Cognitive Impairment (MCI), and 74 with Alzheimer's disease (AD). The statistical information of this dataset is illustrated in Table 5.

**ABIDE**. Autism Brain Imaging Data Exchange (ABIDE) (di Martino et al., 2014; Zheng et al., 2022) is a large-scale autism dataset that provides multi-modal data, including imaging and non-imaging information of the subjects. For a fair comparison, we utilize the Preprocessed Connectomes Project (PCP) (Craddock et al., 2013; Shehzad et al., 2015) to generate four modalities, including Demographic Information (DI), Automated Anatomical Quality Assessment (AAQA), Automated Functional Quality Assessment (AFQA) and FMRI Connection Networks (FMRICN). After pre-processing, this dataset has 871 subjects, consisting of 468 with Normal Control (NC) subjects and 403 with Autism Spectrum Disorder (ASD). The statistical information of this dataset is summarized in Table 6.

### A.2 BINARY CLASSIFICATION.

To evaluate the proposed approach, we further conduct experiments on the TADPOLE dataset. We divide the three categories into three binary classification on the Tadpole dataset, namely, NC and AD, NC and MCI, and MCI and AD. As shown in Figure. 4, the proposed approach has achieved excellent results in terms of ACC and AUC. In terms of NC and MCI, MMGL achieves 96.40% and 94.71%, while our method achieves 97.13% and 96.91% results, respectively. It is worth noting that in the subjects of MCI and AD, the proposed method obtains 97.18% and 94.72% over the methods of comparison, which further confirms the effectiveness of this proposed method. In addition, the variance of our results is relatively small, verifying the stability of our method.

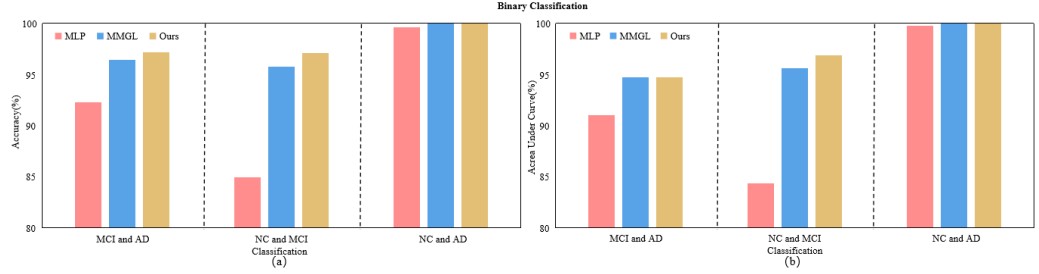

Figure 4: We obtain binary classification results by three methods on TADPOLE.

### A.3 CASE STUDY

To investigate the contribution of each modality in predicting the disease, as suggested in work (Zheng et al., 2022), we adopt the formulation $C_i = \frac{1}{M}\mathbf{1}^T S_i$, where $S_i$ represents inter-modality affinity matrix for the $i$-th subject and $\mathbf{1}$ represents the all-one vector. $M$ denotes the number of the modalities. Through this calculation method, we can understand the degree of dependence between various modalities. As shown in Figure 5, we select 6 subjects from the ABIDE

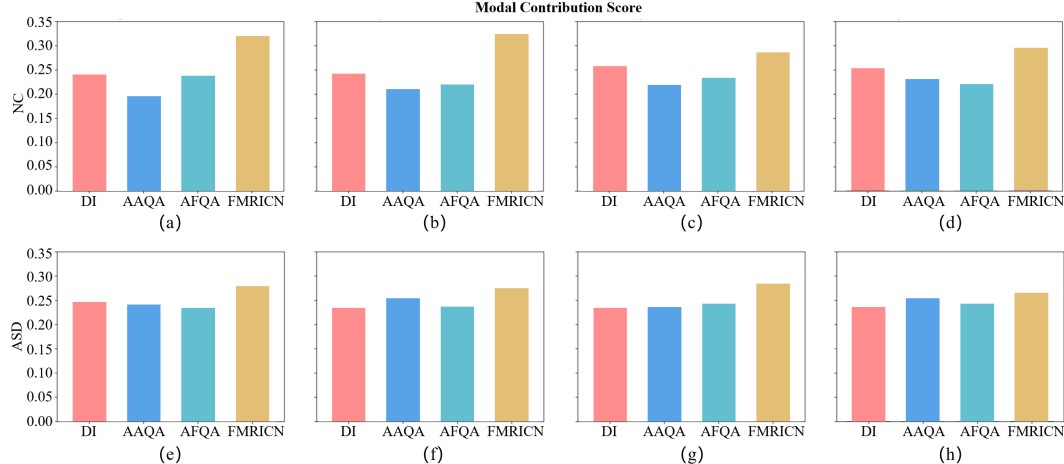

Figure 5: Contribution scores of each modality for six selected subjects, along with the average scores across all subjects. DI: Demographic Information; AAQA: Automated Anatomical Quality Assessment; AFQA: Automated Functional Quality Assessment; and FMRICN: FMRI Connection Networks.

dataset and calculate the proportion of their contribution values. To be specific, it consists of three NCs as shown in Figure 5 (a)-(c) and three ADs as shown in Figure 5 (e)-(g). Then, we calculate the average contribution score of each modality to the subjects in the NC and ASD classes, respectively, as shown in Figure 5 (d) and (h). One can note that imaging modality based on FMRICN plays a key role in disease diagnosis, which is consistent with actual diagnosis.

