# OpenReview forum: "Multi-Modality Brain Disease Prediction with  Progressive Curriculum Graph Learning"
_ICLR.cc/2026/Conference — ICLR 2026 Conference Withdrawn Submission_

### Official Review · Reviewer_PqDj · 2025-10-25

**Soundness:** 2
**Presentation:** 3
**Contribution:** 2
**Rating:** 6
**Confidence:** 3

**Summary:**

This paper proposes a curriculum based learning + multimodal graph based framework (using both feature and label graphs) for brain disease prediction. Experiments demonstrate slight advantages over previous techniques.

**Strengths:**

- Well-specified architecture: dual-relational context graph with learnable feature edges and a label-edge branch, clear loss function equations and modules are concretely presented.

- Progressive multi-modality curriculum pipeline with inter-modality attention is laid out with equations and an explicit label-smoothing curriculum is shown.

- Competitive main results with ablation studies showing benefits of curriculum learning.

**Weaknesses:**

- The paper claims that the task becomes harder with more modalities, but this is not strongly supported. Furthermore, random order vs proposed order of modalities isn't tested.

- Table 2 shows large improvements from MMCL, but LE adds very little gains. Since no statistical significance tests are reported, it's difficult to justify that the label-graph is necessary.

- Despite being claimed as a "general scheme which can be applied for various multi-modality learning with many modalities", only two datasets are tested, and both are within the same domain. To bolster this claim, maybe test on  datasets outside of brain disease prediction.

- This is a general issue I've spotted in this research area: cross-paper comparability is very unclear. While you use 10-fold CV and processed datasets, the paper does not say whether prior fold indices (e.g. MMGL, MGDR) are reused.

**Questions:**

See weaknesses.

Additionally:

- With regard to l71, why do existing graph-based or cross attention approaches suffer from increasing the number of modalities? Is there a difference in computational cost? If so, what is the computational cost of this framework?

- How does the visualisation of T-SNE add evidence? This is a very weak and configuration sensitive indicator. I suggest changing the phrasing to show this is for visualisation purposes only and doesn't necessarily add evidence.

---

### Official Review · Reviewer_zpRB · 2025-11-01

**Soundness:** 2
**Presentation:** 2
**Contribution:** 2
**Rating:** 4
**Confidence:** 4

**Summary:**

This paper introduces a novel framework for multi-modal brain disease prediction, designed to address two persistent challenges in the field: the suboptimal integration of feature- and label-based subject relationships, and the difficulty of fusing an increasing number of data modalities. The authors propose a synergistic approach that combines a "Dynamical Context-Graph Representation" with a "Multi-Modality Curriculum Learning" (MMCL) strategy. The context-graph jointly models relationships between subjects based on their intrinsic data features and semantic class labels, creating a rich substrate for a custom Dual Graph Convolutional Network (DGCN). The core innovation, however, is the MMCL module, which reframes curriculum learning from a progression of easy-to-hard data samples to an easy-to-hard progression of data modalities. The model is trained by progressively increasing the number of modalities from one to many, which is hypothesized to facilitate a more robust fusion of complex, high-dimensional data. The authors validate their method on the public ABIDE and TADPOLE datasets, reporting that it achieves state-of-the-art performance against a comprehensive suite of baseline models.

**Strengths:**

1. The conceptualization of curriculum learning for modality fusion is novel and interesting, from sample-based progression to modality-based progression.
2. The main experiments are comprehensive. The authors validate their framework on two public benchmarks against an extensive suite of eleven baseline methods.

**Weaknesses:**

1. The empirical results do not fully substantiate the claim of achieving the new state-of-the-art performance. The performance gains over the best-performing baseline methods are marginal. Considering the standard deviations, the proposed method seems to achieve performance that is statistically comparable (using t-test) to the state-of-the-art baselines (Table 1).
2. Similar to Table 1, if considering the standard deviations, it’s hard to justify the contributions of each model component (Table 2), or the performance gains of some more complex additions (Tables 3 & 4).
3. The framework defines its "easy-to-hard" curriculum based on the quantity of modalities rather than their intrinsic difficulty, progressing from a single source to many. This design makes the model's performance contingent on the sequential order of modality introduction. Yet, the paper fails to specify the sequence used or analyze the model's sensitivity to this critical variable. This omission represents a significant methodological gap, as the arbitrary choice of the initial modality could create a path dependency that influences the final learned representation and predictive outcome.
4. The description of the training protocol is ambiguous, particularly regarding the implementation of the multi-stage curriculum loss function. Also, there is a lack of hyperparameter sensitivity analysis.
5. The DGCN architecture employs a single set of learnable weight parameters, shared across both feature- and label-propagation branches. This design choice implicitly assumes that the optimal transformation for aggregating neighborhood information is identical for two semantically and structurally distinct graphs: one based on dynamic feature similarity and the other on static class identity. This is a strong constraint that may be suboptimal.
6. The Multi-Modality Curriculum Learning (MMCL) module incorporates a label smoothing schedule where the smoothing factor, $\alpha$, is designed to increase as more modalities are introduced. This hard-codes the assumption that task difficulty invariably increases with the number of modalities. This heuristic may not be universally valid; the addition of a highly discriminative modality could, in fact, simplify the classification task, in which case a less-smoothed, more confident predictive distribution would be more appropriate. The schedule's inflexibility prevents the model from adapting to the actual information content of the fused modalities.

**Questions:**

Please see the Weaknesses section.

---

### Official Review · Reviewer_xuFi · 2025-11-01

**Soundness:** 2
**Presentation:** 3
**Contribution:** 3
**Rating:** 4
**Confidence:** 5

**Summary:**

The authors point out that existing approaches for multi-modal brain data analysis primarily focus on graph construction while overlooking the label perspective. They also note that as the number of modalities increases, learning becomes challenging due to the heightened model complexity.

To overcome these limitations, the proposed method introduces a context-graph, where the adjacency matrix is computed based on feature similarity across modalities and further optimized using a Dirichlet energy function. In addition, another adjacency matrix is constructed to connect nodes that share the same label. Feature representations are then refined through a Dual Graph Convolution Module, which updates modality-specific representations and combines them through summation.

Furthermore, the authors propose a multi-modality curriculum learning strategy that progressively enhances the interaction among different modalities during training.

Through experiments in the TADPOLE and ABIDE datasets, the authors demonstrate the effectiveness of the proposed method.

**Strengths:**

- The authors propose a novel method that enforces smoothness on the graph from both the feature and label perspectives.

- The introduction of curriculum learning is a sound and effective strategy to address the complexity of multi-modal data learning.

**Weaknesses:**

- Despite the methodological complexity, the performance improvement achieved by the proposed approach appears incremental.

- In Figure 2, the performance gap narrows significantly as the labeled ratio decreases. Given that biological labels are typically scarce, this represents a substantial limitation.

- A sensitivity analysis (e.g., on the parameter β) is missing and should be provided to assess the model’s robustness.

**Questions:**

- Since the reported performance improvement is relatively modest, the authors are encouraged to present additional advantages of the method, such as computational efficiency or reduced training time.

- One possible reason for the reduced performance gap at low labeled ratios could be the reliance on the label-based graph. To validate this, the authors should perform an ablation study under the 10% labeled setting by removing the label graph, thereby isolating the contribution of the other components.

---

### Note · Authors · 2025-11-12

I have read and agree with the venue's withdrawal policy on behalf of myself and my co-authors.